# The *Brooklyn Papyrus* Snakebite and Medicinal Treatments' Magico-Religious Context

Wendy Golding

Department of Biblical and Ancient Studies, University of South Africa, Pretoria 0002, South Africa; wgolding@iafrica.com

**Abstract:** This paper investigates the role of magic and religion in the context of the *Brooklyn Papyrus* (47.218.48 and 47.218.85) snakebite treatments. It examines the extent to which these two factors are involved in the treatments and how they shed light on the importance of treating the mind and body of the patient. Information regarding the methods and ingredients used in snakebite treatments in ancient Egypt is obtained from the author's doctoral thesis in which the *Brooklyn Papyrus* (47.218.48 and 47.218.85) is transliterated and then translated into English and offers a commentary on the text. This translation enabled the author to understand that magic and religion form part of the snakebite treatment process. Investigating the relationship between these two factors and medical treatment ingredients and methods enables one to know that magic and religion are inextricably linked in the role of healing. The role played by magic and religion in these treatments resulted in a holistic form of treatment in the process of attempting to heal the ancient Egyptian snakebite victim and patient.

**Keywords:** *Brooklyn Papyrus*; medical papyrus; snakebite treatment; ancient Egyptian snakebite treatment; ancient Egyptian medicine; priest of Serqet; *kherep Serqet*; magic in medicine

## 1. Introduction

The *Brooklyn Papyrus* (47.218.48 and 47.218.85) is an ancient Egyptian manuscript housed in the Brooklyn Museum and is not on display. The papyrus, written in hieratic, is believed to be a copy dating to the Late period (664–332 BCE) that that is based on an earlier original (Sauneron 1989).

At first glance, the treatment section for snakebites, contained in the second part of the *Brooklyn Papyrus*, reads like a recipe book. Essentially, the priests of the goddess Serqet (specialist priests who treated snakebite and scorpion stings) had a scientific approach to snakebite treatment (diagnosis, treatment, and possible outcome) and an initial assessment of the obvious magical and mythological content of the papyrus appears to be less than 10 per cent. 'Magical and mythological content' refers to incantations, invocations, and words addressed directly to the venom.

However, is it possible that the ingredients chosen for the treatments were selected for their perceived or observed therapeutic value as well as for magical and religious associations? A closer look at the links between some of the treatment processes involved in conjunction with medicinal ingredients and magic and mythology suggests that there is a lot more to the simple listing of treatment ingredients and methods of preparation and use than initially meets the eye.

## 2. Background

### 2.1. A Background to the Brooklyn Papyrus (47.218.48 and 47.218.85)

The *Brooklyn Papyrus* arrived at the Brooklyn Museum in 1947, bequeathed by Miss Theodora Wilbour. Her father, Charles Edwin Wilbour (a journalist and Egyptologist), had obtained the papyrus in Egypt and its exact provenance is unknown. When he passed away in 1896 in a hotel room in Paris, the papyrus was among his possessions and was

found in a biscuit box along with the well-known Elephantine papyri (Wilson 1964). It was brought to the attention of Serge Sauneron in the 1960s, and in 1967, and again in 1969 at the Brooklyn Museum, Sauneron worked on conserving the papyrus and transliterating the hieratic script into hieroglyphics, from which he undertook his translation into French. Sauneron was appointed director of the French Archaeological Institute in Cairo in 1969, a position he occupied until his untimely death in 1976 (Leclant 1981; Golding 2020).

The papyrus was a manual for the *Kherepw Serqet* (or the priests of the goddess Serqet) who were called upon to treat those who had been bitten by snakes. The first part of the papyrus describes various snakes to the extent that one can provide probable and possible identifications. It also describes the effects of their bites and the prognosis for the bite victims (Sauneron 1989; Golding 2020). The second part of the papyrus concerning the snakebite treatments is the focus of this article.

*2.2. The Second Part of the Brooklyn Papyrus*

The second part of the papyrus lists approximately 140 snakebite treatments and details their application for snakebites. The treatments cannot be called remedies as we do not know what the treatment outcomes were. These treatments follow a distinct pattern: what the prescription is for (bite of a specific snake or a specific symptom), treatment ingredients, how to put the ingredients together, and how to administer the treatment to the patient. Lastly, a possible outcome for the patient was given. This very much follows current medical treatments: diagnosis, prescription dosage and application, and hopeful outcome. These treatments included poultices and skin medicaments, medications to be swallowed, and emetics (those that were used to induce vomiting) (Golding 2020).

The treatment ingredients can clearly be placed into the following groups: botanical ingredients, animal products, and mineral ingredients. It is useful to provide a fourth category—that of carrier liquids. The ingredients very often needed to be mixed in a liquid of some sort to allow for palatable and easy ingestion, or to bind dry ingredients for a poultice. Sometimes these carrier liquids were used for perceived medicinal benefits of their own (Golding 2020).

Botanical elements make up most of the *Brooklyn Papyrus* treatment ingredients. These botanical ingredients took many forms, from bark, roots, leaves, and flowers to oils, saps, and resins. Occasionally, the treatment did not specify what part of the plant was to be used. The botanical ingredients could be used fresh or in dried form. The most commonly used animal ingredient was honey, followed by blood, excrement, gall, and fat. Products from cattle and goats were frequently used, as were specific species of fish. The most commonly used mineral ingredient in the snakebite treatments was salt. In terms of infection control, this may have been the most useful of all the ingredients. Alum and ochre (both red and yellow ochre) were also used fairly often. The commonly used carrier liquids for the ingredients were wine, water, beer, and sweet beer (Golding 2020).

*2.3. Literature on the Role of Magic and Religion in Healing*

A number of general works have been written about medicine in ancient Egypt and some include chapters on the role of magic in medicine. Ritner (2008), Nunn (2002), and Pinch (2006) discuss the role of magic in medicine but not in relation to the *Brooklyn Papyrus* and its snakebite treatments, although Nunn (2002) says that the *Brooklyn Papyrus* treatments are 'relatively free of magic'. These three authors all discuss the role of cippi and their use in protection against dangerous creatures and the belief that water, poured over a cippus and drunk, will cure the patient from a venous bite or sting. A cippus is a stone stele with an image of the god Horus on it. In one hand, he grasps a scorpion and a dangerous animal of some sort, and in the other hand, he grasps snakes along with another dangerous creature while standing on the backs of crocodiles. There are often inscriptions on these cippi, but they are anti-venom spells rather than treatments.

More recently, Bardinet (2018) has published a work on the role of physicians and magicians at the court of the pharaoh. This study is based on the medical papyrus Louvre E 32847.

Johansson (2019) acknowledges the lack of research into the magical aspects of medical papyrus studies. The focus of studies appears to be more on the 'rational' aspects of the medical papyri. The importance of magic in healing is therefore overlooked. Johansson's work examined the incantations in four medical papyri (Ebers, Edwin Smith, Hearst, and London medical papyri). These medical papyri all date to the 18th dynasty (1550–1295 BCE). However, there are numerous other medical papyri which all have the potential to be examined from a magico-religious perspective.

In Leitz (1999), there are translations of BM EA 9997 and 10309, two papyri that Leitz believes to be from the same document. These two papyri have a very magical approach to snakebite treatment. Another papyrus included by Leitz (1999) in his publication is BM EA 10085, which contains two conjurations that are entirely mythological in nature in addressing snakebite treatment.

The work conducted by Sauneron in the 1960s (and published posthumously in 1989) on the *Brooklyn Papyrus* (47.218.48 and 47.218.85) provides a transliteration and translation into French of the papyrus along with commentary (*Un traité Égyptien d'ophiologie*). There is no commentary specifically dealing with the magico-religious context of the snakebite treatments in this work. Aufrère (2012) discusses snakebite symptoms according to the first part of the *Brooklyn Papyrus* but does not delve into the treatment section in part two of the *Brooklyn Papyrus* in any detail. Brix (2012) also examines the first part of the *Brooklyn Papyrus* and provides a valuable contribution in attempting to identify the snakes described in the first part of the papyrus.

The unpublished doctoral thesis of Golding (2020) on the *Brooklyn Papyrus* (47.218.48 and 47.218.85) provides a transliteration (absent in Sauneron's 1989 publication) and a translation into English. Golding (2020) provides a new commentary on the papyrus.

No work examining the *Brooklyn Papyrus* (47.218.48 and 47.218.85) from a magico-religious perspective has been published.

### 2.4. Magic and Religion in Healing

The ancient Egyptians were great believers in magic. Approaches to medicine and healing must obviously be understood from the perspective of the ancient Egyptians. Our Western way of thinking tends to separate magic and religion into two distinct categories but in ancient Egypt they were very much intertwined and often inseparable (Johansson 2019). This concurs with the comment of Ritner (2008) who says that, as magic was practised on a daily basis in order to maintain balance and order, the ancient Egyptians did not separate the magical from the medical in healing practices.

Magic therefore played a very important role in the healing process in ancient Egypt, as it does to people from many different cultures across the world, both ancient and modern. The World Health Organisation refers to health as a 'state of complete physical, mental, and social well-being' (Mondal and Das 2022). This statement therefore acknowledges the physical and mental aspects of wellness. In treating the individual back from a state of non-wellness, the whole person must be considered.

So important was magic to the ancient Egyptians that they embodied it in a deity whom they named Werat Hekau—a goddess represented in the form of a cobra. Magicians even used snake-shaped wands, which were probably a representation of this very goddess of magic (Pinch 2006).

Although the *Brooklyn Papyrus* snakebite treatment texts have very little obvious magical content, the use of incantations, magical acts, and inferences may well have been performed by the healers in order to appease and placate the patients. Such actions would have provided a placebo effect by calming the mind of the patient, and a calm patient is more receptive to treatment and recovery. According to Nunn (2002), the placebo effect was highly likely to have been relied upon in treating patients, especially if infused with

magic and if incantations were performed. This approach, which relies on the suggestion and expectation of a cure, would have been quite acceptable in ancient Egypt because it is highly likely that there were very few medical treatments that were actually effective. This suggestive placebo method of treatment, incorporating magical acts and words, may have had some value in pain relief (Nunn 2002).

### 3. Magical and Mythological Inferences in the *Brooklyn Papyrus*

On a psychological level, magic had its value in promoting a positive mind-set in the patient. With magic and religion being such an integral part of daily life, it is interesting that less than 10 per cent of the *Brooklyn Papyrus* treatments have obvious magical content, such as recitations, which incorporate invocations, addresses to the venom, or mythological references or allusions.

It was common practice to align the patient with a specific deity through the use of mythological inferences in incantations and as the deity was healed in the incantation or myth, it was believed that the healing would likewise pass to the patient. Treatments that include references to the deities are not exclusive to the *Brooklyn Papyrus*. The *Ebers Papyrus* (a medical papyrus of 110 pages dating to circa 1534 BCE), for example, declares that it has remedies that have been prepared for the gods themselves (paragraphs 242–247) (Nunn 2002).

Incantations could be addressed to the 'disease demon' directly (Nunn 2002) and in this the *Brooklyn Papyrus* was no different. In the *Brooklyn Papyrus* specifically, the recitation is sometimes addressed directly to the venom, commanding it to leave the body of the patient and fall to the ground. An example of this direct address to the venom is found in Paragraph 41 of the *Brooklyn Papyrus*: *tꜣ mtwt! mj! pr ḥr tꜣ!* O Venom (lit. this venom)! Come! Come out upon the ground! (Golding 2020). Similarly, Paragraph 43b holds a direct address to the venom: *mj! prj ḥr tꜣ! jnk ḏḥwtj, smsw sꜣ rꜥ!* Come! Come out upon the ground! I am Thoth, eldest son of Ra! (Golding 2020).

#### 3.1. The Egyptian Word pẖrt

The Egyptian word for treatment, *pẖrt*, is a word with strong magical overtones, where the verb *pẖr* with its possible meanings of 'to go around' and 'to circle' is suggested to be the root of the word *pẖrt* (Ritner 2008). The treatment is intended to 'encircle' and protect the patient. This 'encircling' of the patient can be seen in the treatments that are made to cover the body of the patient in six of the treatments contained in the *Brooklyn Papyrus*. Two examples follow here:

Paragraph 43c, *Brooklyn Papyrus*, page 3, lines 6 to 7

*kt: nnwt nt ẖt n ẖf(ꜣw) jn.tw ḥr dšrt jꜣbty. nḏ snꜥꜥ ḥr jrp s[gn]n nḏmt, r-pw, swr jn wḥd dmt,ḥnꜥ nḏ snꜥꜥ ꜥḥmw.f ḥr bꜣq. nḏ ẖꜥw n wḥd dmt jm. m šs mꜣꜥ(t). ṯsj ḥ(ꜣ)ty, srq ḥtyt. jw jr.tw.f r dr nsy(t) mjtt.*

Another (remedy): roots of wood-of-the-snake (plant) (which) one brings from the Eastern Desert. Crush smoothly in wine or sweet [ointment], whichever. To be swallowed (lit. drunk) by the one suffering the bite, and crush its leaves and stems finely in moringa oil. Cover and protect (lit. cover) the body of the one suffering the bite with it. (It is) truly excellent. The heart rate elevates, (and) the throat can breathe. One shall use it to drive out epilepsy and the like. (Golding 2020)

Paragraph 66b, *Brooklyn Papyrus*, page 4, line 15

*kt: snṯr wꜣḏ, ḥmꜣt mḥt, mrḥt, bjt. nḏ snꜥꜥ m ẖt wꜥt. nḏ.s jm.*

Another (remedy); fresh terebinth, salt of the north, goose fat, honey. Crush smoothly into a homogenous mixture (lit. 'into one thing'). Cover and protect (lit. cover) the person (lit. him) with it. (Golding 2020)

There is much more involved in a patient's treatment (*phrt*) than the simple act of putting ingredients together and treating the patient according to instruction. Treating a patient is a magical act.

### 3.2. The Use of Magical Actions in Treatment

Reading through the snakebite treatments of the *Brooklyn Papyrus* reveals that certain actions associated with magic are performed in some of these treatments. These actions were equally as important as the ingredients used. The magical actions were performed because it was believed that they had a purpose that was vital for the success of the treatment. These actions included recitations over the ingredients, drawing, making fumigations, spitting, and swallowing. A treatment was thought to become more effective when words were recited over the ingredients because it was believed that they would become infused with *heka* (magic) (Pinch 2006).

### 3.2.1. Drawing

Other than fumigation, one of the first examples of magical action to be found in the *Brooklyn Papyrus* occurs in Paragraph 79c, a paragraph that contains specific instructions for the incantation in the preceding paragraph (Paragraph 79b). The incantation was recited over images of the deities Ptah, Isis, and Serqet whose images were drawn on a piece of new papyrus and placed on the patient's throat. As Pinch (2006) says, it is important to understand that an image or even a name could represent the real thing. The drawing of these deities could have been intended to represent their physical presence at the healing session.

### 3.2.2. Spitting

According to Ritner (2008), the act of spitting is an important magical action. It is found in Paragraph 84 of the *Brooklyn Papyrus* in which the patient must spit onto the ground four times. It is reminiscent of the commands to the venom in Paragraphs 41, 43b, and 80b of the *Brooklyn Papyrus* in which the venom is addressed directly and ordered to 'fall out upon the ground', leaving the patient's body.

One of the intended results of the magical act of spitting is purification, and the hope that the power of the spitting act will cure or resuscitate the patient, because the spittle is a 'conveyer or medium' for the 'blessing or the curse' (Ritner 2008). An interesting method of spitting involves filling one's mouth with water or other liquid and pouring it onto the head of the patient. This same concept of purification by spitting is used in the *Brooklyn Papyrus* in an attempt to purify the body by removing the harmful venom.

Not only is the actual act of spitting of value but spittle itself is of great importance because, says Ritner (2008), it was used in creation in ancient Egyptian mythology. Spittle is present in the myth of Isis and Ra in which Ra's spittle is used by Isis to create a snake which bites him, and this myth, according to Ritner (2008), is a spell for healing scorpion stings. In Section 3.3.2 below, it shall be shown how this very same myth was referred to in the *Brooklyn Papyrus* as an equally important part of snakebite treatments.

### 3.2.3. Swallowing

Ritner (2008) presents swallowing as another magical action. A person needs to swallow in order to receive nourishment. The same act internalises treatment on a magical level; for example, in the *Pyramid Texts*, the missing Eye of Horus is often identified with food offerings. The eating and subsequent swallowing of the offerings internalises the Eye of Horus and symbolises a return to the body and, presumably, wholeness.

There are a number of examples of the patients needing to swallow treatment mixtures in the *Brooklyn Papyrus*. Additionally, there is a treatment in Paragraph 85c, in which the healer or the Serqet priest must recite the words 'I know it', while removing a piece of broken-off fang from the bite wound. The fang must be expelled into a bowl of beer and dates and this mixture must then be drunk by the bite victim.

3.2.4. Recitations over Treatments

Incantations appear to be common in medical treatment but not in trauma (Nunn 2002). In a few of the *Brooklyn Papyrus* treatments, words were pronounced over the ingredients in the hope that the treatment would harness the magical powers of the words. These words and their magic would then be transferred to the snakebite victim either in a drinkable treatment or into the bite wound in the instances where the words are pronounced over a poultice treatment, or even transferred by massage into the limbs of the patient. Paragraph 80a of the *Brooklyn Papyrus* (page 5, lines 8 to 9) instructs the healer to grind natron and mix with vinegar and an unknown ingredient and apply this under a bandage to the patient's bite wound. Before the mixture is applied, however, a recitation is said over it. The instructions for this are in Paragraph 80b (page 5, lines 9 to 13):

> *šd n.s: m.tn bṯt jwtj ꜥnḫwj.f, pr m jm ḥwt-nṯr.f, r.f wn sꜣw.f m p dp, dbḥw rḏw jnpw, r djt n.f sw jm [...ḏd] jn ḥr n ẖr(j) dmt.f: m.k! wj ḥr sqd jmj r.k! jr khb.k wj, (jw.j) m sꜣ.k! jr psḥ, jw.j ḥr šd.f. dr.n.j mtwt pw r wꜣt. jr psḥ ḥr, wr šnw.f. mtwt! [pr] n sj n st, mj psḥ.k m ḥm.k. jw wsjr, ẖftjwb.f ḥr! ḏd js jr bṯt nn ꜥnḫwj.f, ḥnpw pw. ḥr tw r.s ms n msw-bdš, jqꜣšꜣrw. wn mwt.[j] srḳt ḥr nḥm ꜥnḫwj.f. j(w)s ḥr ḥtm r.f r mdt. šp.k bṯt, ḥnp! nn sdm r.s,*
>
> *r djt n.j pr mtwt.s ḥr tꜣ. nn sš.s m ꜥt nb n sj n st. nn sqd.s. m ꜥt.f nb. nn ẖdb.s m jwf.f.*

Pronounce over this remedy (lit. 'it'): See the betjet which does not have ears, coming out of his temple/tomb where he stays, in order for him to act as guardian in Pe and Dep, which is necessary for the bodily excretions (lit. fluids) of Anubis, in order to place him in/on him in [. . .]. Horus [saying] to the one who has the bite wound: Look! I am making which is in your mouth turn around! If you harm me, (I will) be following you! As for the bite, I chase it away. I have driven the venom out of your body (lit.'to the path'). As for the bite of Horus, great is his magician. Venom! [Come out] of X, son of Y (lit. 'a man of a woman'), as you bite without being known (lit. 'in ignorance of you'). Osiris, his enemies fail. Say of the betjet without ears, 'It is a henep'. One calls it the child (young) of the mesou-bedesh, Iqasharu. It is [my] mother Serqet who removed its ears. She seals its mouth to prevent speech (lit. 'against speech'). Be blind, Betjet, Henep! Its voice is not heard, in order to make its venom fall out upon the ground. It does not spread in any limb of X, son of Y. It will not circulate in any of his limbs. It will not cause death to his body (lit. 'kill in his flesh'). (Golding 2020)

Paragraph 80c of the *Brooklyn Papyrus* (page 5, line 13) then instructs the healer to use some of this mixture and massage the patient:

> *ḏd ḥr pẖrt tn. sjn ꜥwt n wn dmt jm.s m ꜥwj.k, ḥnꜥ kꜣp.f.*

Spell to be recited over this remedy. Rub the limbs of the one who has been bitten with the remedy (lit. 'it') with both your hands, and fumigate him. (Golding 2020)

In Paragraph 98c of the *Brooklyn Papyrus* (page 6, lines 17 to 19), magical words are said over myrrh which is then burnt and the patient is fumigated with the smoke. Another example involving magical words and fumigation is found in Paragraph 99c of the Brooklyn Papyrus (page 6, lines 24 to 28):

> *kⸯrⸯ(t): jsw pwy nn pr m mnw, ... pr [...]rḥwj, pr n.j. smꜣ n.s ḥr. js šw.f. sw m ꜥ.f, jm m ꜥ. s mꜣ.k tꜣ mt[wt ...]. rr (pẖr) r ḥrt. ḥꜣ ḥr r jb n ẖftj n wsjrÚ dj.k ntr nb ḥr mnw [snb ... ḏd mdw] ḥr ḥkr ḥbs, rdjt ḥr nḥp jsw wḏ(w). ntš [...]. kꜣp.s(j) ḥr.s n snb.f.*

Another (magical spell); O these reeds which come from Min, go out [. . .] Two Companions, go out for me! Horus, make (the venom) ineffective for him. It is dried out. It is there in his hand. You will destroy this ven[om . . .]? Turn around, (come) from the sky, and come down upon the heart of the enemy of Osiris! May you cause every god who is suffering [to heal . . . Say these magical spells] over

decorative items and cloth. (For) placing on a potter's wheel (with) green reeds. Sprinkle (with water) [. . .]. Fumigate the person with it until he recovers his health. (Golding 2020)

*3.3. The Use of Mythology in Invocations*

The complexity of treatment becomes apparent in the *Brooklyn Papyrus* with the use of invocations and incantations that have roots in mythology. Some of the myths are unfortunately no longer extant. Those myths that are extant can give us insight into the religious and magical facet of healing and medicine in ancient Egypt.

In an invocation, a deity is called upon or appealed to for assistance in the healing of a patient. Misfortune and illness were believed to be bestowed on individuals by the deities (Nunn 2002) and consequently it is they who can restore health. In the *Brooklyn Papyrus*, the deities that are associated with magic and healing, such as Serqet, Isis, Horus, Ptah, and Thoth, are referred to in the paragraphs of magical treatments. However, only Horus and Thoth are appealed to directly for assistance.

An example of invoking Horus is found in Paragraph 99c of the *Brooklyn Papyrus* (see Section 3.2.4 above).

Thoth is invoked in Paragraph 43b of the *Brooklyn Papyrus* (page 3, lines 2 to 6):

*ḏd ḥr.f m ḥkꜣw. jw ḏḥwtj ꜥpr m ḥkꜣw.f, ḏbꜣw m ꜣḫw.f r šnt tꜣ mtwt.*

*nn sḫm.t m ꜥt nb(t) n s n st, mj šnt sbjw m-sꜣ sb.sn ḥr rꜥ ḏs.f.*

*ḫnr.k sw m ḥꜥ nb n s n s, mj ḫnr.k tꜣwj n rꜥ. ḫꜥm mꜣꜥt r šnbt.k jsw jry. ḫꜥ.k r.s, ntr pwy špss, sꜣ ntrt wrt ḥkꜣw. šnt.k s n s, mj šnt.k mn ḏs.k hrw pwy n pgꜣ qꜥḥt.k. sḥr.k sw r tꜣ m ꜥt nb n s n s, mj sḥr.k sbjw twy sbj ḥr wsjr. sḥr.k tꜣ mtwt r r n psḥÚ*

*m.k jn.n ntr ḫt jm.f ḏs.f r sḫt, r drt, r bḥn tꜣ mtwt nt ḥf(ꜣw) nbt ḥf(ꜣt) nbt ntj mḥ ꜥt nb(t) n s n st.*

*mjÚ prj ḥr tꜣÚ jnk ḏḥwtj, smsw sꜣ rꜥÚ*

Recite over him with magic spells. Thoth comes, equipped with his magic, (and) equipped (lit. adorned) with his magical power to exorcise this venom.

You will not have power over any limb of X, son of Y (lit. a man of a woman), (just as the insurgents were exorcised after they rebelled against Ra himself.

You shall lock it out from all the flesh of X, son of Y, (just) as you lock up the Two Lands (Egypt) of Ra. Maat approaches your breast (as) a substitute thereof. You appear before the venom (lit. it), O Noble God, son of the Great Goddess of Magic! You shall exorcise X, son of Y, like you exorcised your own suffering, on the day of the wounding of your shoulder.

You shall throw the venom (lit. it) down to the ground from every limb of X, son of Y, (just) as you overthrew the rebels that rebelled against Osiris. You shall make this venom fall from the opening (lit. mouth) of the bite wound!

Behold! The god has brought something divine (to put) in the wound (lit. itself) to make this venom fall, and to expel (it), (and) to remove (this venom) of every male snake (and) of every female snake, which fills every limb of X, son of Y.

Come! Come out upon the ground!

I am Thoth, eldest son of Ra! (Golding 2020)

There are two important ancient Egyptian myths in which a god is bitten by a snake and subsequently healed. References to these two myths appear in some of the recitations in the *Brooklyn Papyrus*. These mythological references would, no doubt, provide reassurance to the snakebite victim and hopefully induce a positive state of mind in that he/she, too, would be healed. It is helpful to summarise these two myths for the benefit of the reader who is not familiar with Egyptian mythology.

### 3.3.1. The Myth of Horus and Isis

The snakebite myth involving the deities Horus and Isis is a part of the Osiris myth. In this myth, Isis gives birth to Horus and conceals him in the reeds while she goes on a visit to the city of Am. Returning to the reeds, she finds Horus very close to death. She cries for help and the people who live in the papyrus swamps come rushing to her aid, but nobody has the knowledge needed to resuscitate Horus. At first it was thought that Seth, brother of Horus, had caused this demise but it was then discovered that Horus had been bitten by a snake (in some versions of the myth Horus is stung by a scorpion). The goddess Nephthys arrives and grieves with her sister Isis. She is accompanied by Serqet who enquires of Isis what happened to Horus. Nephthys tells Isis to call out to the heavens for aid from the gods. Thoth hears the cry and arrives with his magical powers. Horus is healed and he returns to life (Pinch 2006; Nunn 2002).

As a result of being healed, Horus appears as a conqueror of dangerous animals in the Metternich Stele of the Thirtieth Dynasty (393–343 BCE) and the cippi of the Late Period (664–332 BCE). As one who has survived the sting of a scorpion and the bite of a venomous snake, it is as if Horus has immunity from them.

In the *Brooklyn Papyrus*, a treatment with references to this particular myth of Horus and Isis is found above in treatment 43b (*Brooklyn Papyrus*, page 3, lines 2 to 6).

The importance of appealing to Thoth can be understood in context when one knows the myth of Horus and Isis. It is not surprising that Thoth, as the patron god of medicine, is called upon because he is the god in the myth who comes to save Horus from the scorpion sting or snakebite, bringing with him his magic powers and spells. In Paragraph 43b of the *Brooklyn Papyrus*, Thoth, as god of magic (among other things), is requested to exorcise the venom and throw it to the ground. Thoth played an important role in magic and medicine because not only could he write, he could also recite the spells that were written down (Nunn 2002).

### 3.3.2. The Myth of Isis and Ra (or, Ra's Secret Name)

This myth comes from the *Turin Magical Papyrus* (Pinch 2006) and there are references to it in the Papyrus Chester Beatty (a medical papyrus) (Nunn 2002). In the myth, Ra, the creator god, has many names and these names are not even known by the other deities. Isis believes that if she knows Ra's sacred name it would give her power to become the supreme goddess of the earth and enable her to have a status in the heavens like that of Ra. Ra is growing old and his saliva runs from his mouth and falls to the ground. Isis gathers together soil and the great god's saliva which she forms into a snake. She leaves this snake on a path frequented by Ra and, as she expected, Ra does not see the snake and it bites him. It is a venomous bite and Ra begins to suffer and life starts to slip from him. As he is filled with pain and sickness, Ra calls upon all the gods. Isis, too, arrives and she bring along her magical powers and words. Isis tells Ra that he has been bitten by a snake and that she can heal him. She asks Ra for his sacred secret name in exchange for healing him. Initially, Ra does not comply but as he feels the ravages wrought by the venom and his life being extinguished, he reveals his name to her (Pinch 2006; Nunn 2002). Isis then heals him, saying: 'Depart, poison, go forth from Ra. O Eye of Horus, go forth from the god, and shine outside his mouth. It is I who work; it is I who make to fall down upon the earth the vanquished poison' (Budge 1971, p. 141).

The treatment in the *Brooklyn Papyrus* that hints at this particular myth is found in Paragraph 79b (*Brooklyn Papyrus*, page 5, lines 3 to 7): [*psḥ.n wj*] *bṯt nn mꜣꜣ.(j). ḏdb.n wj ḥt nn mꜥ.j m ꜥḥꜣw jn šnj r.j. m.k, js bꜣg.kw*! The betjet (snake) [has bitten me] (but) I did not see it. Something has pricked me (which) I did not see. (Golding 2020).

That people in ancient Egypt believed in supernatural forces and that disease or illness could be brought about and subsequently healed by the gods or demons is very apparent in the myth of Isis and Ra. The myth demonstrates the reason why deities were invoked to help with healing. The deity believed to be responsible for the misfortune wrought upon another deity or a person is the one who holds the key to the cure.

This direct address to the venom by Isis at the end of the myth of Isis and Ra to 'fall down upon the ground' appears in a few of the *Brooklyn Papyrus* treatments, and is well-illustrated in Paragraph 41 of the *Brooklyn Papyrus* in an invocation to the onion.

3.3.3. Invocation to the Onion

In the *Brooklyn Papyrus*, it is not only the deities but also plants that are requested to assist in the fight against venom and are invoked just as much as the deities. The most frequently used botanical ingredient in the Brooklyn Papyrus pharmacopoeia is the onion (*Allium cepa*), and the best example of appealing to the onion is found in Paragraph 41 of the *Brooklyn Papyrus* (page 2, lines 19 to 26). It is a rather lengthy example and it may well have an original basis in a myth that is no longer extant so unfortunately we do not know what the myth is. In this Paragraph, the use of the hieroglyphic determinative for divinity ⟨glyph⟩ (falcon on a post) with the hieroglyphics for the word onion (*ḥḏw*) suggests that the onion is viewed here as divine, or from a divine source. The divine status of the onion, therefore, makes it a vitally important and powerful ingredient to include in treatments and invocations. In Paragraph 41, it is clear that the onion is held in great esteem and is greeted in the recitation and referred to as the 'unrivalled one which guards all the gods.' In Paragraph 42a (*Brooklyn Papyrus*, page 2 line 26 to page 3, line 1), it is described as *wnn.f m ꜥ ḫrp srqt* (being under the hand of the Controller of Serqet)—in other words, always readily available.

Paragraph 41 begins with the instructions to the *kherep Serqet* (healer) who then begins addressing the venom when he says 'It is Ra who wards off venom':

> A very good remedy to prepare for someone who has been bitten: onions: grind finely in beer. (Drink) and vomit, for one day. Recite over him with magic spells: a mouth, a mouth, a tooth against teeth. It is Ra who wards off venom. While the mouth of the god is at the place of your mouth, his word will strike down your venom, (and) you will flow out from where you are (lit. the place). O Venom (lit. this venom)! Come! Come out upon the ground! I have brought a tooth in my hand in order to drive you out. This tooth of the great god has been brought. It fell upon the ground (and) by falling back, became youthfully vigorous again, taking root (lit. growing) on the ground, (and) growing green upon the desert path, so that you may be overthrown, (and) in order to strike down your mouth (lit. the place of), (and) to overthrow the marks (lit. position) of your teeth.

Then, the healer addresses the onion:

> Hail to you, Onion! Hail to you, tooth of the god! Hail to you, original/important (lit. foremost) tooth of Osiris! Hail to you, guardian who protects all the gods, by means of your name, that of Onion! May you enter into the stomach of X, son of Y (lit. a person of a person). Ward off all the venom which is in there, by means of your name, that of Onion! You have destroyed that which is in the hand of Ra. Kill that which is in the hand of Horus, (and) that which is in the hand of Seth, (and) that which is in the hand of the Great Ennead of gods. Destroy their enemies there! Destroy their chief for me by means of your name, that of Onion.

> Open your mouth against their mouth, in your name, of 'that which opens the mouth'. Consume them there in your name, of 'that which [consumes]'. Grind their flesh in your name, that of ['tooth']. You destroy this venom, Powerful One, which is in the mind, which is in the heart, which is in the spleen, which is in the liver, which is (in) the lung, which is in the throat, which is in the head, which is in the posterior, which is in every limb of X, son of Y (lit. someone of someone). The heat from the blast of your fire is against the venom (lit.it) in order to kill it. It dies of your bite. (Golding 2020)

This paragraph demonstrates that the healers realised that the physical symptoms associated with an envenomed bite were due to the venom that was transferred from the

snake via its fangs into the victim and that it was thus something that was present in the body of the patient that could not be seen, and needed to be extracted somehow, whether by exorcising or vomiting. The most frequent use of the onion in the *Brooklyn Papyrus* is, in fact, as an emetic to induce vomiting and usually when the bite was from a venomous snake. The reference to a tooth is important here. It was common in ancient cultures to set like against like, for example: fight fire with fire. Here, the onion (which is compared visually to a tooth) is pitted against the fang of the snake.

In Paragraph 41 of the *Brooklyn Papyrus*, the milk tooth of the Great God falls upon the ground and grows green on the desert floor. In this regard, the tooth of the god is likened to the onion bulb, which is white, but grows green leaves once it is planted. Further on in this same Paragraph 41, it is said that the onion protects Horus from those who attend on Seth (i.e., those who do the bidding of the god Seth, both a brother and adversary of the god Horus in Egyptian mythology). This comparison of an onion with a tooth is not unusual. It is found in two utterances for the Opening of the Mouth Ceremony in the *Pyramid Texts* where the onions are referred to as the white teeth of Horus.

In the last line of the recitation, the onion is likened to a snake when dealing with the venom: 'the burn of your flames is against it and it dies of your bite'.

The high esteem that the onion is held in causes one to wonder if other botanical ingredients in the snakebite treatments possibly had mythological or magical associations.

### 3.4. The Link between the Brooklyn Papyrus Medicinal Ingredients and Mythology

An investigation into the link between treatment ingredients in the *Brooklyn Papyrus* and magic and mythology produced some interesting results. A number of these ingredients, especially botanical ingredients, are associated with deities who were bringers of life such as the all-important solar deities, or even creator deities such as Khnum. Some of the plants are linked to resurrection through their association with Osiris, and yet others are connected to the deities of healing such as Isis, Thoth, Hathor, and Horus.

#### 3.4.1. Botanical Ingredients

The link between the onion and mythology is clear in Section 3.3.3 above. This is the most valued plant ingredient in the healer's pharmacopoeia. In addition, there are some trees, whose parts are incorporated into the recipes, which are important to the ancient Egyptians for their protective reputation as a result of their link to certain deities. These are the *Maerua crassifolia* (*jmꜣ* [meru tree]), *Ziziphus spina Christi* (*nbs* [jujube]), the *Salix* species (*trt* [willow]), the *Vachellia nilotica* (*šnḏt* [*Acacia nilotica*, Nile acacia]), and the *Vachellia tortilis* (*ksbt* [*Acacia tortilis*, umbrella thorn acacia]).

Substances of 'mysterious origin', such as resin, were believed to be imbued with *heka*, the essence of magic (Pinch 2006) and, due to their golden colour, were associated with the life-giving sun god. The ingredient known as *šꜣkr* in the *Brooklyn Papyrus* could well be identified as amber (Golding 2020), which is a fossilised resin and a perfect example of this golden colour. According to Serpico (2000), there are no known deposits of amber in Egypt so the ancient Egyptians would have obtained it elsewhere or through trade. A story told by Nicias, an Athenian politician and aristocrat, that amber was formed from the rays of the sun links this ingredient to the deities upon whom the creation and maintenance of life depended, the all-important solar gods.

The acacias (*Vachellia* and *Senegalia* species) were linked to powerful deities associated with rebirth and healing, as well as protection. These are qualities that would be considered vitally important in an ingredient used for healing purposes and so the acacia species appear frequently in the *Brooklyn Papyrus* snakebite treatments. The umbrella thorn acacia (*Vachellia tortilis*) was known as *ksby* while the Nile acacia (*Vachellia nilotica*) was named *šnḏt* (Golding 2020).

Through the association of the acacia with the gods Horus and Osiris, these trees are linked to the processes of healing and rebirth. Horus is linked to the acacia in the *Pyramid Text* 436a–b (Sethe's edition), in which he is referred to as 'Horus who comes forth from the

acacia' (Buhl 1947), and in one of the many Horus myths he took refuge under an acacia when he was a child (Buhl 1947). An inscription in the Edfu temple states that the acacia was the sacred tree of Herakleopolis (capital of the 20th nome of Upper Egypt, located near modern-day Beni Suef). The acacia was planted on the nome's sacred mound, which was believed to be a symbolic burial place of Osiris and the green growth of the tree was considered to imitate the god's rebirth process (Díaz-Iglesias Llanos 2017).

The Nile acacia (*šnḏt*) is a thorny tree with solid wood and so became a symbol of the protection of the king against his enemies. This protective quality was further supported by the acacia's link to the protective goddesses Isis, Nephthys, and Sekhmet (Díaz-Iglesias Llanos 2017). The goddess Sekhmet was also known as 'lady of the Acacia' (Hart 2005). She was also the patron deity of the priestly healers known as *waeb Sakhmet* and, as a bringer of disease, it was also believed that she could heal, hence her epithet 'lady of life' (Hart 2005). The creator goddess Neith was also linked to the acacia, and a temple of the acacia of Neith was located near Crocodopolis (modern-day Fayoum) at the lake of $bddw$-$k_3$ (lit. lake of the watermelon) (Baum 1988).

The protective qualities of the acacia are further seen in the link between the umbrella thorn acacia (*Vachellia tortilis*) and the god Sopdu in the *Pyramid Text* 436a–b and 994b–d, where he is referred to as 'Sopdu who lives under his *ksbt* tree' (Buhl 1947). Sopdu was often depicted with his hand raised in a threatening gesture to drive off supernatural forces and, in his falcon form, he lived in a sacred grove (Pinch 2002). Hart (2005) describes Sopdu as a 'border-patrol god' and, in addition, he was a protector of the Sinai turquoise mines. In the *Pyramid Texts* (456a), the *ksbt* tree was also linked to Sobek, a god credited with the creation of the Nile (Buhl 1947).

The *ksbt* tree must have been held in high regard to be the recipient of offerings. During the Late Period (664–332 BCE), in a procession at the temple of Isis at Philae, the king made an offering to the goddess of the third nome of Nubia with charcoal of *ksbt* wood (Baum 1988). The acacia was not only important enough to receive offerings, but also to have its own sanctuaries. The goddess Sekhmet was worshipped at the Sanctuary of the Acacia in Heliopolis during the Old Kingdom period (2675–2130 BCE) and it was believed that a second sanctuary of the acacia existed at El-Kab in Upper Egypt. Additionally, we learn from a hieratic papyrus from Tebtynis of a Temple of the Acacia of Neith in the Fayoum (Baum 1988).

The *ksbt* tree was the sacred tree of Min in his role of god of the desert and of Coptos (modern-day Qift) from the Middle Kingdom period (2030–1650 BCE) onwards. On a stele (*Lyon E 328*), which is possibly from Abydos, Min of Coptos (Gebtu) received the shade of a stylised tree which could be either *ksbt* or *jm₃* (*Maerua crassifolia*) (Baum 1988). A hymn dedicated to Min-Amun on the *Parme 178 stele*, which dates to the end of the Middle Kingdom period and which is probably inspired in part by *Pyramid Texts*, refers to the god of the people of the *ksbt*. This text also appears at the temple of Seti I at Abydos, and again at the temples of Edfu and Hibis (Baum 1988). The *ksbt* tree is closely linked to the protective divinities of Coptos and Akhmin, which rule in the desert and its routes—a region in which the *ksbt* tree grows and the people worship Min, and also Sopdu, who is worshipped in the eastern parts of Egypt where the tree is found on the eastern border of the Delta, the desert isthmus, and a part of the Sinai (Baum 1988).

Barley *(Hordeum vulgare)* and wheat *(Triticum turgidum, Triticum dicoccum)*, known as *jt* and *bdt* or *mjmj*, respectively, were plants that symbolised resurrection due to their links with the god Osiris (Hart 2005), and, accordingly, necklaces of barley were often placed on mummies (Darby et al. 1977). The concept of grain representing resurrection is understood as follows: the crop is cut down, trampled, and winnowed to release the grain during the harvest time. When the grain germinates and grows to create the new crop it is linked to renewal and resurrection. This process is represented in the myth involving the death, dismemberment, and renewal of Osiris (Pinch 2002). During annual festivals, Osiris figurines were made, filled with mud and grain, and the seeds were sown and watered until they germinated. These were called 'corn mummies' (Pinch 2002) or 'Osiris beds' or

'grain-Osiris' (Wilkinson 2003). The Osiris bed could be covered with linen and then placed in the tomb with the deceased (Darby et al. 1977).

According to Dioscorides (a Greek botanist and physician born *circa* 40 AD), the ancient Egyptians believed that the goddess Isis discovered wild wheat and barley and that Osiris showed humankind how it should be cultivated (Täckholm and Täckholm 1941). Barley and wheat were also linked to the god Hapy, who embodied the inundation of the Nile and its life-giving qualities. 'The maker of barley and wheat' was one of Hapy's epithets. Consequently, the flooding Nile revived Osiris who returned with barley as a result of the annual inundation of the Nile (Pinch 2002).

These links between barley and wheat and the life-giving deities, such as Isis and Hapy and the god of resurrection, Osiris, made the grains ideal ingredients to use in snakebite treatments. Barley was primarily used for making beer in ancient Egypt but could also be used for bread-making. In the *Brooklyn Papyrus* treatments, barley is used in a variety of forms. Barley flour and the husks or bran were used exclusively for wound treatments. Liquid from barley mash (perhaps barley water) was used in an emetic and also in a wound treatment. Mucilage of barley was used in treatments to be ingested and also in a wound treatment, while old bread grain was used in an ingestible treatment (Golding 2020).

The botanical ingredient named *snw pt* in the *Brooklyn Papyrus* may be the blue lotus *(Nymphaea caerulea)*. If this is the case, then *snw pt* has important mythological links to regeneration and creation. In *The Contendings of Horus and Seth*, there is a myth in which an angry Horus beheads his mother when she spares the life of Seth (Pinch 2002). In order to punish Horus, Seth tears out both of his eyes and buries them on a mountainside. They emerge as lotus flowers when they begin to grow (Pinch 2002) and they 'light up the earth' (Rundle Clark 1959). In another version of the myth, cited by Darby et al. (1977), the eye sockets of Horus sprout lotus plants which are eventually transformed into eyes. Pinch (2002) reminds us that the lotus is also linked to the sun god who emerges as an infant on a lotus in order to start creation. This emergence of the sun god in a lotus flower forms part of the Hermopolitan creation mythology (David 1980). As a symbol of the sun, the lotus also appears in Heliopolitan mythology where the creator god Atum was believed to have emerged from a lotus bloom (Darby et al. 1977). Another myth involving the lotus tells how the bloom rose up out of the 'Sea of Two Knives'. The lotus opened to reveal a scarab which transformed into a child which cried tears, from which humankind was formed (Darby et al. 1977). In yet another myth, the god Nefertem—who was originally a lotus flower—is described as the 'lotus at the nose of Ra' and he is shown in anthropomorphic form with a lotus upon his head (Buhl 1947). The blue lotus is the lotus associated with these myths, and it became a symbol for rebirth, as is suggested by funerary art in which the deceased would be depicted holding a fragrant lotus bloom to his or her nose. The scent of the blue lotus was believed to bestow new life upon the deceased as a 'follower of Ra' (Pinch 2002).

Cilician fir *(Abies cilicica)* produces oil *(sft)* that has been identified as one of the seven sacred oils. Botanical oils were important because the pharaohs made offerings of them in the temples to the deities by the pharaohs. The embalming process used ten sacred oils (Roth 2003).

The date palm *(Phoenix dactylifera)* is yet another plant linked to healing and resurrection through the deities Hathor and Osiris. The goddess Hathor had her home in the third Lower Egyptian nome (a territorial division in ancient Egypt), Kom el-Hisn, where her local name was *nbt jmȝw* (mistress of the date palms). The ancient Egyptian word *jmȝw* was the name given to the male date palms, while the more familiar name *bnr* referred to the female trees (Buhl 1947). Trees played an important role in the Osirian myths, and it was believed that the date palm grew from the innards of Osiris (Baum 1988). Dates *(bnw)* used in snakebite treatments therefore had associations with the two very important deities, Hathor and Osiris.

The fig *(Ficus carica)*, *dȝbw* in ancient Egyptian, is depicted in texts and tomb reliefs and considered to be a divine food, according to the *Pyramid Texts.* The pharaoh Ramesses III gave an offering of 15,500 measures of figs to the god Amun-Ra (Darby et al. 1977).

The grape vine *(Vitis vinifera)* has long been associated with deities and mythology in several ancient cultures. As early as Egypt's Old Kingdom Period (2675–2130 BCE), lists of offerings included wine and grapes (Baum 1988). A myth from *the Papyrus Jumilhac* illustrates a link between wine offered to the gods in the temples and the Eye of Horus. In this myth, Anubis, the god of embalming and care of the deceased, takes boxes containing the eyes of Horus and buries them on a mountainside. Here, they are watered by the goddess Isis in order to bring them back to life, and they eventually emerge as the first grape vines (Pinch 2002). In addition, not only were grapes considered to be the Eye of Horus, but the resulting wine was considered to be the tears of the Eye (Baum 1988). The Eye of Horus is of exceptional importance here as it was believed to be imbued with magical protective powers. In Paragraph 41 of the *Brooklyn Papyrus*, the Eye of Horus is invoked to destroy the venom in the body of the patient.

Quantities were often expressed hieroglyphically as parts of the Eye of Horus and these symbols are used extensively in the *Brooklyn Papyrus* treatments as well as in other medical papyri.

Each part of the Eye  represents a fraction of the *heqat*—a unit of measurement approximately 4.5 ℓ—as follows

$^1/_2$ 
$^1/_4$ 
$^1/_8$ 
$^1/_{16}$ 
$^1/_{32}$ 

$^1/_{64}$ 

Grapes, as the Eye of Horus, expressed a return to natural order in funerary rites, and they were a symbol of physical integrity and balance in magical texts where they were eaten or wine was drunk (Baum 1988). The connection between resurrection and the grapevine is confirmed by the illustrations covering many tomb ceilings of vines and bunches of grapes (Baum 1988). Wine *(jrp)*, the product of grapes, as a carrier liquid in the *Brooklyn Papyrus* snakebite treatments therefore represents a manner of transferring treatment ingredients into the body of the patient that is imbued with qualities of powerful magic and resurrection. Raisins *(wnš)* were also used in the treatments and would have been considered to carry the same qualities (Golding 2020).

The jujube, or Christ-thorn *(Ziziphus spina-Christi)*, was regarded enough in ancient Egypt to have its own cult. A sacred grove of jujube *(nbs)* trees grew at the provincial frontier town of Saft el Henne (ancient Per-soped) in the delta region (Baum 1988) where the main cult of the *nbs* tree was located. It was called *ḥt nbs,* or the House of the Ziziphus tree (Buhl 1947). During the Graeco-Roman period (332 BC–395 AD), there was a temple at Dakke (located in Lower Nubia) that was named 'house of the *nbs* tree' (Buhl 1947). So important was this tree that it was considered to be a sacred tree in no less than 14 nomes (Buhl 1947), and a number of towns in the nomes of Heracleopolis, Hermopolis, and Antaeolopois bore the name *ḥwt-nbs* (Baum 1988). Various deities such as Ra-Horakhty, Hathor, Shu, and Tefnut were shaded by the *nbs* tree (Baum 1988) and, along with the meru tree *(Maerua crassifolia)*—named *jmꜣ* in Egyptian—served the deceased in *Pyramid Text* 8081a–b (Buhl 1947).

The meru tree *(Maerua crassifolia)* appears as a sacred tree in several Egyptian nomes. This tree (called *jmꜣ* in the ancient Egyptian language) became the sacred tree of the fertility and harvest god Min from the end of the New Kingdom Period (1550–1069 BCE) (Baum 1988). In the great nome list of Edfu, the *jmꜣ* is found alongside the *ksbt* tree which was the sacred tree of the fifth nome of Upper Egypt in the Ptolemaic era. At Dendera, the *jmꜣ* is one of the sacred trees of the sixth nome listed in the Graeco-Roman temple of Hathor (Baum 1988). The naming of trees as sacred trees of various territories in ancient Egypt indicates their importance.

The moringa tree (*Moringa pterigosperma*, *Moringa aptera*, or *Moringa peregrina*, *Moringa oleifera*) is associated with the creator god Ptah and the ancient tree god Kherybaqef. Additionally, the moringa is linked to the god of magic, Thoth, who, as Thoth-Kherybaqef, resides in the chapel of Nefertoum at the temple of Seti I at Abydos (Baum 1988). From the end of the Third Intermediate Period (1075–656 BCE), Kherybaqef appears on sarcophagi as a funerary deity incorporated into the Four Sons of Horus who protect the deceased (Baum 1988). The oil of the moringa was considered to be the Eye of Horus in the 'opening of the mouth' ceremony and the tree itself was highly regarded because it assisted in the continuation of the cycles of nature, assuring the victory of the eye of the god in the sky and also the passage of time from one year to the next. Apparently, on the first epagomenal day, the priest who represented the king travelled in a palanquin of moringa wood to the chapel where a ceremony intended to divert all danger from the renewal of the year and also from the king's reign was performed (Baum 1988).

Reeds obtained significance through the mythological role they played in the establishment of the first temple. In the beginning, there was no life and no deities. From the dark waters, a mud island began to emerge and a reed settled upon its shores. A few demi-gods came into being and they rescued this reed by planting it into the soil. Soon, the first god, a falcon, settled upon it. As more land emerged, more reeds were added and so a shelter began to be created. The shelter grew bigger until eventually it became a temple. The land immediately around it was considered to be sacred because of the falcon that alighted upon the first reed (David 1980).

There is not much to be found linking the watermelon *(Citrullus lanatus)*, a plant that has been present in Egypt since pre-Dynastic times (5500–3100 BCE), to mythology or the gods, save for the fact that the ancient Egyptians believed that the watermelon (*bddw-kꜣ*) grew from the semen of a frustrated Seth, which was scattered on the ground when he transformed himself into a bull while pursuing the goddess Isis (Manniche 1989). The germinating seeds link the watermelon to rejuvenation (Golding 2020).

Symbolically, the willow *(Salix safsaf* or *Salix mucronata, Salix aegyptica)*, known as *trt* in ancient Egyptian, was closely linked to Osiris as it was believed to provide shelter for his dead body. Many tombs and towns had willow groves associated with them and a 'raising the willow' festival was held annually as an agricultural fertility ritual (Wilkinson 1992). These sacred groves were established in towns to house the *ba* (an aspect of the soul of the deceased) of Osiris (Buhl 1947). It was believed that the *bennu* bird in its incarnation of the soul of Ra alighted on the willow tree in Heliopolis (Buhl 1947). It was also believed that the bird landed in the *trt* tree at the onset of creation and that various deities including Ra-Horakhty, Horus of Edfu, and Osiris took on an aspect of this bird (Baum 1988).

### 3.4.2. Animal Ingredients

As with the botanical ingredients in the *Brooklyn Papyrus* snakebite treatments, it is worth considering if the animal products were selected for use in the treatments for their perceived magical properties or even chosen because the animal had some desirable characteristic. Several animals whose by-products were used in the snakebite treatments have links to mythology, pharaohs, and deities.

Colour was very important to the ancient Egyptians and each colour had significant meanings. It has been noted in the treatments in the *Brooklyn Papyrus* that occasionally the colour of an animal, from which blood, organs, or excrement is obtained, is specified, for example: fat of a black cow or blood of a red goat. Black was a popular colour in magic and it was often specified that blood or milk must be obtained from a black animal. Although the colour red was often associated with negativity such as chaos and evil, it was also linked to the solar eye goddess and, because of this link, was also considered to be highly powerful (Pinch 2006).

Ingredients derived from cattle would have been very powerful ingredients to include in treatments due to the strong link between these bovine creatures and several important deities. Consider the perceived characteristics of the wild bull, such as virility, strength,

and aggression, which were often attributed to the Egyptian pharaohs, who frequently took part in wild bull hunting, not only to display these desired characteristics but also as a demonstration of order, represented by the pharaoh succeeding over chaos—the chaotic world being represented by the wild bull in the hunt (Darby et al. 1977). In fact, the inclusion of cattle names into the Egyptian nomes of the north of the country prior to the unification of ancient Egypt strongly suggests that cattle were already playing a strong role in the religious and secular lives of the human population (Darby et al. 1977). After the unification of Egypt (*circa* 3100 BCE), the worship of cattle began to spread from the north through the rest of the country.

Cows were considered to be sacred during the Dynastic Period (3100–30 BCE) and were linked to the goddess Hathor in her role as mother of the sun god Ra and they were also sometimes associated with the goddesses Isis and Nut (Darby et al. 1977). It was during the Dynastic Period that, according to a third-century CE writer named Porphyry, Egyptians did not eat meat from a cow as it was considered sacred (Darby et al. 1977). The well-known ancient Egyptian cult of the sacred Apis bull is believed to have been introduced during the Old Kingdom Period (Darby et al. 1977). From the New Kingdom Period onwards, the mummified bulls worshipped in Memphis were interred in the Serapeum at Saqqara.

Blood, the ingredient known as *snf*, is frequently used in the *Brooklyn Papyrus* snakebite treatments. Together, with the mineral ingredient known as *djdj*, blood has links to the goddess Hathor and the creator sun god Ra in the *Myth of the Destruction of Mankind*. In this myth, the goddess Hathor sets out to destroy humankind and the god Ra must stop her. He calls upon his messengers to collect large quantities of a red mineral (presumably Nubian or red ochre). This mineral (*djdj*) is ground up while maidservants make beer from barley mash which they then mix with the mineral. This colours the beer red so that it looks like a field of blood. The Eye Goddess, in the form of Hathor, sees her reflection in this red beer and is so delighted that she forgets that her intended mission was to destroy humankind (Pinch 2002).

Donkey hair, knotted to make amulets, had a magical status and medical treatments made use of their fat, liver, blood, gonads, and bone marrow (Darby et al. 1977). None of these ingredients are used in the *Brooklyn Papyrus* snakebite treatments but donkey hoof and excrement are used in wound treatments in the *Brooklyn Papyrus*. As other donkey products had a magical status, it is possible that donkey hoof and excrement were likewise awarded the same status (Golding 2020).

The gazelle (*Dama schaeferi*), a currently endangered species found in the Sahara, appears in many Dynastic Period representations of hunting scenes, and was often offered in sacrifice by pharaohs. A myth tells how, at the very first Sed Jubilee (an important ceremony which was performed to celebrate the continued reign of a pharaoh) of the pharaoh Mentuhotep IV, a gazelle offered herself as a sacrifice (Darby et al. 1977). Gazelle (Egyptian *hnn*) horn, liver, heart, and blood were all used in the *Brooklyn Papyrus* snakebite treatments (Golding 2020).

The hippopotamus was associated with the protective goddess Ta-urt (Darby et al. 1977). In the *Brooklyn Papyrus*, one finds hippopotamus fat used in a wound treatment.

Honey was probably one of the most useful ingredients in the snakebite treatments in terms of wound infection control. Honey is known to be a natural antibiotic. The bee became a symbol for Lower Egypt which was an especially good honey production area due to its delta agricultural lands (Darby et al. 1977). There were even officials employed to keep bees and collect honey which was an important offering to the fertility god Min (Darby et al. 1977). Honey was also known as 'liquid gold' and linked to the sun god Ra whose tears, in one particular myth, turn into bees (Darby et al. 1977). Any ingredient linked to the all-important sun god Ra was bound to be imbued with value and therefore considered to be a good ingredient for medicinal use.

Milk was used in offerings to the gods by the pharaohs. Numerous reliefs show the goddesses Isis and Hathor suckling an infant Horus (Darby et al. 1977) and, as a highly valued food source, milk and its by-products were also medicinal ingredients and therefore

appear in the *Brooklyn Papyrus* snakebite treatments. Curds would have been made from the milk of cows, sheep, and goats (Serpico and Whyte 2000). The high regard in which milk was held is illustrated by its appearance in a few ancient Egyptian myths. In the *Contendings of Horus and Seth*, Seth removes the eyes of Horus and the goddess Hathor cares for the injury to the eyes by using the milk (El Saeed 2016) of a gazelle (Rundle Clark 1959). In another myth, at the base of the trees at the Abaton (one of the alleged tombs of Osiris at Philae), milk was poured in order to 'revive and rejuvenate' Osiris at the moment of his rebirth (Buhl 1947). In yet another myth, depicted on a statue of the pharaoh Amenhotep II (*circa* 1440 BCE), the pharaoh is shown drinking from the udder of Hathor. In the myth, the deceased Amenhotep meets Hathor while on his way to the Underworld. She gives him milk from her udder thus bringing him back to life. The use of milk as a reviving substance in these myths enables one to understand how important its use was in medicine.

### 3.4.3. Mineral Ingredients

Although a variety of mineral ingredients were used in the *Brooklyn Papyrus* snakebite treatments, very few had links with mythology and magic. The most commonly used mineral ingredient was salt and both red and yellow ochre were also used fairly frequently (Golding 2020).

A red ochre called *djdj* is possibly a special type of red ochre found in Aswan (Elephantine) and Nubia (Golding 2020). This ingredient is linked with the goddess Hathor in the *Myth of the Destruction of Humankind* (see 'blood' in Section 3.4.2 above) which is found in *The Book of the Heavenly Cow*. Clay was linked with creation and life. The ancient Egyptians believed that the god Khnum created humankind from clay or mud on the potter's wheel (Pinch 2002, p. 68). As a creator god, Khnum provided an important link with the giving of life and, as clay was associated with him, it would have been a strong ingredient to use.

### 4. Conclusions

Magic and medicine in ancient Egypt were complimentary skills. This is evident in the *Brooklyn Papyrus*, a unique treatise on snakebite treatment that takes into account both the physical and mental aspects of attempting to heal the snakebite victim. Treatment in the *Brooklyn Papyrus* occurred on a physical level (treatment of the body) as well as on a psychological level (treatment of the mind, including emotional and spiritual aspects), and can be thought of as holistic.

Certain actions performed in the *Brooklyn Papyrus* treatments (spitting, swallowing, drawing, fumigating, recitations) were believed to have magical associations. Just as important were the invocations and incantations. The patient would have been reassured by these words as they were pronounced over treatments. He or she would hopefully have been calmed by the incantations and the invocations to the various deities or the commands to the venom to leave the body.

As Nunn (2002) comments, certain ancient Egyptian myths form the basis of understanding the medications, amulets, and incantations used in healing. This is certainly true of the *Brooklyn Papyrus.* Invocations in the *Brooklyn Papyrus* often had a basis in mythology and inferences to myths were made during the treatment process.

While a number of ingredients in the *Brooklyn Papyrus* snakebite treatments have links to various deities and mythology, plants in particular are linked to deities of the sun, creation, healing, and protection. The overwhelming number of treatment ingredients were of a botanical origin. Certain trees had sanctuaries dedicated to them, had festivals in their honour, or received offerings. Others were the sacred trees of nomes (Egyptian provinces) or had sacred groves established near towns. One plant, the onion, was considered to be of divine origin.

There appears to be sufficient evidence to suggest that a number of the ingredients selected for the treatment of snakebite victims in the *Brooklyn Papyrus* could have been chosen on the basis of their associations with mythology and important deities rather than selected for their perceived or observed therapeutic values.

Overall, the *Brooklyn Papyrus* (47.218.48 and 47.218.85) is neither completely magical in its approach, nor is it completely rational (dealing with medical treatments only). The document presents a valuable opportunity to understand how magic, religion, and medicine all work together in order to provide a balanced attitude towards treatment, and to learn how this was approached by the ancient Egyptian *Kherep Serqet*.

**Funding:** This research received no external funding.

**Institutional Review Board Statement:** Not applicable.

**Informed Consent Statement:** Not applicable.

**Data Availability Statement:** No new data were created.

**Conflicts of Interest:** The author declares no conflict of interest.

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
