# Peer review of "The Brooklyn Papyrus Snakebite and Medicinal Treatments’ Magico-Religious Context"

_religions, doi:10.3390/rel14101300_

Round 1
Reviewer 1 Report
This may depend also on the editorial concept of the volume, but sometimes the author's approach is far too essayistic for a research paper. For example, the author says: "The ancient Egyptians were great believers in magic, which formed an integral part of daily life. Our modern-day thoughts and opinions cannot be imposed on the ancient mind and we must seek to understand approaches to medicine and healing from the perspective of the ancient Egyptians". This is firstly superfluous, secondly, from the standpoint of methodology also unusual - of course, we are not imposing our standards on ancient cultures.
The bibliography is also a bit outdated - except for Golding work (2020), all other references are at least ten years old, not to mention Budge.
Lines 214-215 The author mentions "version of the snakebite myth involving the deities Horus and Isis can be found in Budge (1971)." Guess we are talking about one of the episodes of the Osiris myth? It should be said more clearly and with the proper bibliographic references.
Author Response
I have attached a revised manuscript and response to reviewers

Reviewer 2 Report
Overall this is a well-written and well-argued paper that presents original data (new translation), analyses and contextualizes the ancient text, and demonstrates the validity of the argument. As stated, it comes from a PhD thesis on the same topic. I have a few comments intended to strengthen the overall manuscript, and I suspect much of what I mention below is actually in the thesis but didn't make it into this paper. I reassure the author that this is a common theme among junior scholars and easily remedied.
Having read through the paper, the conclusion is a bit abrupt. Having aptly demonstrated that most of the ingredients and actions prescribed in the cures have associations with magic, it might be good to flesh the conclusion out a little and explain why it is important (to Egyptology) that we now know this. There is more meat in the introduction than in the conclusion, and they should match.
My primary concern is that the paper seems somewhat under-referenced and most of the references are more than ten years old, although with Egyptology some of that is unavoidable given the long history of publications on the subject. Still, only three references are less than ten years old. One of them, Golding 2020, seems to be another thesis on the Brooklyn Papyrus. It might be advisable to mention this thesis in the text and specify what new contributions this work produced. The two citations leading to it both seem to relate to possible origin of ingredients. Abundant references to Ritner 2008 are appropriate, as Robert's work is generally appreciated to be central to modern Egyptology (and his premature death a great loss).
Basically, the paper lacks a literature review. It would be helpful to have not just a historical account of the papyrus's journey from Egypt to Brooklyn (which is very well done but lacks any citations whatsoever) but also an account of how it has been interpreted and discussed by other scholars, if at all. There are citations to other works on Egyptian healing and magic throughout, but not a discussion of the place of this papyrus in the scholarship. Has it been frequently used and cited, or largely ignored? Situating both the manuscript and the central argument of the paper within current Egyptology would be very helpful. Is the argument that 'most if not all ingredients and methods of medicine had magical meaning' a generally accepted idea, or does it go against the flow of current thinking? Who else has been working on this topic lately, and do you agree with their interpretations?
The other topic on which the paper is under-referenced is holistic healing. This is a fascinating topic with a lot of recent literature in anthropology, psychology, and medical sources. There are no sources related to holistic medicine cited in the paper. Statements about the calming affect of rituals and incantations on patients (101-102 and 104) are unreferenced. There are so many interesting discoveries about the mind-body connection that it would be good to see at least a reference to current literature, especially when the subject is brought up but not referenced. As stated in the paper, we cannot know the efficacy of any of these treatments, only to observe that some ingredients seem much more likely to be beneficial than others.
There is a type-o on line 413. The text reads 'an offering of to the goddess of the third nome of Nubia...' There is a word missing between 'of' and 'to,' or perhaps the word 'of' was meant to be deleted. The subject of the sentence is the ksbt tree. Other than that, the English is excellent and no errors were noted.
Author Response

(The authors gave the same response as above.)

Reviewer 3 Report
Your article is very informative and thus, suits a wider audience with little prior knowledge of ancient Egyptian medicine and magic.
In the attached PDF, I have highlighted passages whose language needs attention because the wording is awkward or the grammar is weak.
In addition, some highlighted passages are followed by a note, and in those case, I have made comments on the ideas and arguments you are attempting to convey.
My general observations are:
- define early on what you mean by Egyptian "magic" and how you contrast it to "medicine/healing" (be careful of the risk with our modern bias which you, too, have brought up at some point)
- citations without page references are to me incomplete, but this could be the result of the journal's stylistic guidelines
- several of your arguments are weak, because you do not provide concrete examples from texts or artifacts
- you seem to rely much on popularized overviews of Egyptian mythology and magic, such as Pinch and Manniche's books: this, in combination with the shortage in concrete examples from the ancient materials, weakens considerably the strength of your arguments
- you are using "importance" as logical justification for many of your arguments, but it is not enough to convince an expert audience
- be consistent in your manner in which you present passages from the Brooklyn papyrus: always include your transliteration (it seems that there is an issue with transliteration fonts) and translation
- some of the transitions from one section to another are rather abrupt (see e.g. 3.3.3 to 3.4)

I have highlighted in the attached file all passages whose language needs attention.
Author Response
I have attached a revised cover letter and response to reviewers

Round 2
Reviewer 3 Report
Thanks for revising it. The article's updated version is much more refined and your arguments are articulated and supported in clearer manner.